# Epidemiologic and Epizootic Data of Tularemia in the Past and in the Recent History in Croatia

**DOI:** 10.3390/microorganisms8050721

**Published:** 2020-05-12

**Authors:** Mirna Mihelčić, Valentina Marečić, Mateja Ožanič, Ina Kelava, Maša Knežević, Marina Šantić

**Affiliations:** Faculty of Medicine, University of Rijeka, Brace Branchetta 20, 51 000 Rijeka, Croatia; mirna.mihelcic@medri.uniri.hr (M.M.); valentina.marecic@medri.uniri.hr (V.M.); mateja.ozanic@medri.uniri.hr (M.O.); ina.kelava@medri.uniri.hr (I.K.); masa.knezevic@medri.uniri.hr (M.K.)

**Keywords:** *Francisella*, tularemia, Croatia, endemic region

## Abstract

Tularemia is a zoonotic disease caused by *Francisella tularensis*. A large number of recent studies have provided an update on the disease characteristics and the distribution across Europe. In Croatia, most of the clinical cases, as well as the reports of the disease in animals, date from the 20th century. In that period, epidemic and epizootic research had given detailed information about endemic regions and their characteristics, including suspected animal hosts and vectors. The region along the middle course of the Sava River, called Middle Posavina, is described as an endemic region, i.e., a “natural focus” of tularemia, in Croatia. In the 21st century, cases of human tularemia are being reported sporadically, with ulceloglandular, oropharyngeal and typhoid forms of disease. A majority of the described cases are linked with the consumption of contaminated food or water. The disease outbreaks still occur in areas along the course of the river Sava and in northwest Croatia. In this review article, we have summarized epidemiologic and epizootic data of tularemia in the past and in recent Croatian history.

## 1. Introduction

*Francisella tularensis* subsp *holarctica* (Type B) is spread all over the northern hemisphere [1] and it is the most often-detected subspecies in Europe. Assigned to category A agents according to Centers for Disease Control and Prevention (CDC) [2,3,4], *Francisella tularensis* poses a huge threat to humans and animals. Disease is transmitted by direct contact with infected animals or vectors, or indirectly. Indirect transmission includes consumption of food or water contaminated by animal excrement or carcasses. Consumption of contaminated water or food is the most frequently reported infection route in Europe. Then follow contact with infected animals and arthropod bites [5]. Clinical manifestations of tularemia in humans depend on route of transmission. If transfer occurs by inoculation of bacteria via arthropod vector, glandular and ulceloglandular forms of the disease appear. After ingestion of bacteria by contaminated food or water, oropharyngeal tularemia develops. Other forms of the disease include the occuloglandular form and, the most severe, typhoid [3,6,7,8,9]. Tularemia is present in many European countries, with either sporadic cases or outbreaks. Sweden and Finland have the highest notification rates among EU countries in recent years. However, in Czech Republic, Germany and Norway, human tularemia cases have also been frequently reported. Iceland, Ireland, Cyprus, Greece, Latvia, Luxembourg, North Macedonia, Malta and the United Kingdom are described as countries free of tularemia [8,10]. Based on the accessible data, the Republic of Croatia is still a country with a low incidence of tularemia but sporadic occurrence [10]. Little is known about *Francisella’s* ecology and its dispersal in Croatia. New knowledge is necessary to complete the research achievements from the past.

While investigating tularemia in Croatia, its geographical location should definitely be taken into consideration. Croatia is located in the northwestern part of the Balkan peninsula, and is geographically diverse, comprising several ecological regions (Figure 2). The coastline is the Mediterranean part of the country, the mountains are a part of the Dinaric Alps, the flat plains are part of the Panonian plain, situated between three huge rivers: Drava, Danube and Sava. Concerning its ecological characteristics, the continental part of Croatia can be compared to a floodplain forest–meadow ecosystem, which has been associated with the ecology of *Francisella* in Central Europe. In the central and the western part of the European continent, the terrestrial lifecycle of *Francisella’s* species has been described. Direct contact with infected animals and tick bites are presumed to be the predominant routes of transmission. Terrestrial mammals, especially lagomorphs and rodents, are considered as a potential reservoir of *Francisella* infections [11,12,13,14]. *Francisella* has been isolated in Europe from different tick genera including *Haemaphysalis concinna*, *Dermacentor reticulatus* and *Ixodes ricinus* [15,16,17,18]. In the region of South Moravia in the Czech Republic, the presence of *F. tularensis* has been associated with the floodplain forest–meadow ecosystem along river courses. Occasional inundations result in an outbreak of tularemia in this endemic region where *F. tularensis* subsp. *holarctica* has been isolated from ticks belonging to genus *Ixoides* [19]. The ulceroglandular form of the disease in humans has been described as common in Slovakia, Austria, Czech Republic and Germany [11,12]. In addition, vole species, such as field vole, bank vole and common vole, have been reported as potential vectors and reservoirs of tularemia in Germany and Switzerland [20]. Kuehn and colleagues reported positive serum samples on *F. tularensis* subsp *holarctica* obtained from foxes, raccoon dogs and wild boars captured in Germany [21]. Concerning the disease’s seasonality, it is driven by a transmission route and the ecological characteristics of the bacteria in different parts of Europe. In Germany, the majority of human cases are reported during summer and autumn months. The cause of this seasonal pattern is primarily related to increased outdoor activities of the population, and thereby, a possible contact with the wildlife [22].

Nevertheless, the aquatic cycle is also present in Europe, and the main transmission route is the ingestion of contaminated water from streams, ponds, lakes and rivers [10]. This cycle dominates in the southernmost part of Europe. Waterborne outbreaks have been described in Turkey, Kosovo, Bulgaria, Georgia, Northern Macedonia and Italy. However, in Germany, the aquatic cycle has also been reported [23]. In Turkey, the consumption of contaminated water is a prevalent mode of *F. tularensis* transmission and the oropharyngeal form of tularemia is the most common in humans. The majority of tularemia outbreaks occur in the autumn–winter period [24]. Two large outbreaks were confirmed in Kosovo between 1999 and 2002. Outbreaks have been connected with consumption of contaminated food and water and poor sanitary conditions in the postwar period. Oropharyngeal manifestation of the disease predominated [25,26]. Due to the frequency of reported outbreaks of waterborne tularemia in countries neighbouring Croatia, located in the southeast of Europe (Kosovo, Bulgaria, North Macedonia, Turkey), the existence of an aquatic cycle cannot be excluded even on the Croatian geographic territory.

Various phylogenetic analyses of *F. tularensis* subsp. *holarctica* isolated in Europe have sought to confirm the hypothesis about the dispersal of the pathogen from east to west in the recent past. Little genetic variation and a low genetic diversity among *F. tularensis* subsp. *holartctica* strains simplified monitoring of bacterial spread in the past. The first reported outbreaks in western European countries date back to the first half of the 20th century, when outbreaks occurred in Austria, former Czechoslovakia, Poland, Germany and France [27]. Based on the phylogenic analysis, four main genetic clades and appropriate subclades are confirmed: B.4, B.6, B.12 and B.16 [8]. Most of the European isolates belong to the phylogenetic groups B6 and B12 (Figure 1). Each of the clade and additional subclades are predominantly associated with a specific geographical distribution. Phylogenetic analyses show close genetic relationships among *F. tularensis* isolates on the west part of the European continent. Isolates belong to the Branch 11, which matches subclade B.Br. FTNF002-00, comprising B.45 and B.46 clades. Subclade B. Br. FTNF002-00 has been reported along with France, Spain, Italy, Netherlands, Germany and Switzerland [28,29,30,31], while B.12 clade and its subclade B.13 are mostly distributed in Central, Eastern Europe and the European part of Russia. On the border between the European and Asian continents, in the territory of Georgia, the majority of isolates confirmed the B. Br.013 group [32]. The B13 subclade has been identified in Germany and Switzerland as well [22,29,31]. In the Asian part of Turkey, clades B.6, B.12 and B.16 have been reported [8,33,34]. Therefore, the authors proposed that the spread in the west could occur by expansion of the specific population originating from the east part of the European continent [27].

Whether Croatia was included during the *Francisella* dispersal from eastern European boundaries to the west is yet to be determined. Concrete data for the *F. tularensis* lifecycle in Croatia are still missing or are incomplete, and a comprehensive study is necessary, primarily from genetic approaches. Despite the limited data available, this manuscript seeks to summarize a comprehensive overview concerning former knowledge about tularemia in Croatia.

## 2. Epidemic Investigations of Tularemia in Croatia—Efforts to Date 

The first report of human tularemia in Croatia dates back to 1952, when the disease occurred in the middle stretches along the Sava river. It is a large area between the capital city of Zagreb and the town of Slavonski Brod (Figure 2). In the second half of the 19th century, five epidemic outbreaks have been described: 1952–1954, 1964–1965, 1967–1969, 1974 and 1998. The majority of the registered cases were reported along the middle reaches of the Sava river [36].

At the same time, the interest for investigation of this zoonosis had risen among scientists, physicians and veterinarians. During the tularemia outbreak between 1952 and 1954, over fifty human tularemia cases were registered and the disease was confirmed by isolation and cultivation of *F. tularensis* from clinical samples. All described cases occurred after direct contact with infected hares (*Lepus europeus*) [37,38]. It was an intensive period for investigation of tularemia in Croatia during which Croatian scientists adapted and established an immunological intracutaneous tularin test. Initially, it had been developed by Foshay in the early 1930s [39]. Although with some differences in the preparation of the allergen, the test was widely implemented in the USA and in the former Soviet Union. Tularin is an allergen, a protein component of an inactivated francisella. After the intracutaneous administration of tularin, previously infected persons or persons vaccinated with live vaccine for the experiment developed hypersensitivity. The test is highly specific and stays positive for a lifetime. To improve the allergic test, Croatian scientists had developed a new method of producing the allergen for the tularin test. The refined test was characterized by a higher specificity and a simpler production method. In brief, *F. tularensis*, strain 76, isolated from mouse *Apodemus agrarius* in 1965 near the town of Sisak was used. Bacterial cells were destroyed by repeated freezing and thawing. The suspension was then filtrated, and the soluble antigen was titrated by a standard serum to 1:4 and autoclaved. Tularin was then applied intracutaneously in the forearm and the reaction was observed 48 hours after the administration. The occurrence of induration of seven or more millimetres together with erythema and edema, was considered positive [40].

In 1964 and 1965, when tularemia re-emerged in the area around Sisak (Figure 2), Heneberg and his group explained the epidemiological characteristic of the disease, through the implementation of the tularin test. Based on anamnestic data and questionnaires, 15 cases of tularemia were reported. Contact with hares was the main source of the infection. For retrograde detection of tularemia cases among the inhabitants, which had not been reported previously in the designated area, the tularin test was performed. From 623 healthy persons involved in the survey, 27 of them (4.3%) were positive on the tularin test. According to the authors, they might have become infected with tularemia in one of the previous outbreaks and remained positive on the tularin test. [38]. In the autumn of 1967, a tularemia outbreak re-emerged in area region around the town of Vrbovec (Figure 2), situated northeast of Zagreb. Until spring 1968, the outbreak spread south, alongside the Sava river. Altogether, 58 clinical cases of tularemia were confirmed. About 60% of people with clinical signs of the disease reported previous contact with hares. Moreover, 1926 people were tested with tularin and 75 were positive. Anamnestic data revealed that more than a half of them had one febrile episode during the outbreak. In parallel, the agglutination titers on *Francisella* in sera were confirmed positive for both groups, clinical tularemia cases and tularin-positive persons [41]. Repeated outbreaks of the disease and a lot of confirmed cases of tularemia in a relatively small geographical area, in villages near the Sava valley, directed further investigation studies to certain ecosystems.

Borčić and his group, at the end of the sixties and the early seventies of the 20th century, performed further epidemiological surveys. The main goal of the study was determination of a “natural foci” of tularemia in Croatia, i.e., areas where the recurrence of the disease can be further expected. Borčić describes tularemia as a “lowland disease” that is closely related to water ecosystems. Thereby, the main emphasis was given to the localities situated in the main lowland and flood plain areas nearby large Croatian rivers. The survey involved villages and towns near great rivers including the Sava, Drava, Danube, Kupa, and Gacka. The characterization of the natural foci also included the determination of the sources of human infection and the transmission route. Altogether, 58,962 persons were involved in the survey. Most of them were schoolchildren and youths. The tularin test and the serological agglutination test were performed, together with a clinical examination and questionnaires. The tularin test was positive in 137 persons. Interestingly, 118 positive persons lived in region of the middle course of the Sava river, and most of them did not recall any contact with hares. Overall, collected data enabled the researchers to define the three main natural foci of tularemia in Croatia: the greatest one was localized southeast of Zagreb, along the left bank of the middle course of the Sava river, in the region of Middle Posavina. Two other smaller areas were in northeastern Croatia, Međimurje and around Koprivnica, near the Drava river (Figure 2) [37]. In addition, the authors described the seasonal dynamics of tularemia. It started in autumn, reaching its peak during the winter and declining during the summer time [42]. The predominant clinical forms were ulceloglandular and glandular tularemia [43,44].

In 1974, the outbreak of tularemia affected a large geographic area, including the middle courses of the rivers Sava and Kupa. From 32 registered cases, 29% revealed the typhoidal and 20% the oropharingeal form of tularemia. All patients suspected of tularemia revealed a positive tularin skin test and a positive agglutination titer in sera. The presumed reservoirs of infection were hares, but the contact with them was confirmed in just a few cases [36].

In the past twenty-five years, tularemia in Croatia occurs sporadically in humans. Although, there are no sustained epidemiologic studies, few documented case reports of human tularemia describe different transmission routes. In 1996, Mišić-Majerus, described four human cases of tularemia transmitted by vectors. Anamnestic data of one 37-year-old male patient and a nine-year-old boy revealed tick-borne transmission. The other two were a 49-year-old female and her 12-year-old son. They both confirmed mosquitoes and other unidentified flies bit them while they were fishing. All patients lived in the area close to the Drava river (Figure 2). Patients suffered from the ulceroglandular form of the disease and were positive for the tularin test [45].

Another tularemia outbreak in Croatia in recent history occurred at the end of 1998 and the beginning of 1999, again in the endemic area. The outbreak numbered 18 clinical cases of tularemia and emerged in Petrinja, a town near the Kupa river (Figure 2). The majority of the infected persons revealed an oropharyngeal form of disease. The suspected source of the infection was contaminated food, but the consumption of hare meat or any contact with them could not be proven. The tularemia outbreak occurred shortly after the Croatian War of Independence. Tularemia outbreaks in the postwar period occurred by contamination of food in storage houses due to low sanitary conditions and increases of small rodent populations [46].

In 2011, Grgić and colleagues described a case report of a 43-year-old male with a typhoid form of tularemia. The disease manifested with remittent fever, and tularemia was confirmed serologically. The epidemiological data reveal previous contact with water contaminated by small rodents and dead animals, during the construction of the Rijeka–Zagreb motorway [47]. Finally, in 2016, Puljiz and colleagues described one sporadic tularemia case from 2015 in a 44-year-old female patient who suffered from oropharyngeal tularemia. The disease was confirmed serologically. This tularemia case was linked with consumption of contaminated water and food according to epidemiological data [48]. Altogether, data from European Centre for Disease Control and Prevention (ECDC), revealed more than twenty registered cases of tularemia between 2012 and 2018, with the peak at 2015, when 13 cases were confirmed in Croatia (Table 1) [49,50].

According to the directive prescribed by the government of Croatia, the highest incidence in human tularemia cases still occurs in the continental part of the land, i.e., in areas along the course of the Sava river and in northwest Croatia [51]. The Croatian Agency of Agriculture and Food reported one tularemia outbreak, transmitted by food, in 2015, with five confirmed cases reported to the Croatian Institute of Public Health [52]. In addition, two sporadic cases of tularemia occurred in 2016, and three in 2017 [53,54]. Nevertheless, detailed epidemiological data about recent human tularemia cases in Croatia, such as description of the disease, source of infection or described geographic region, have not been provided.

## 3. Epizooty of Tularemia

Most of the data describing the reservoirs of *Francisella* in the nature dates from the 1960s and 1970s. Although hares were determined as the supposed source of infection in the epidemic outbreak of 1964–1965, small rodents were under the investigation loop and were hallmarked as potential reservoirs of tularemia in nature. Therefore, almost 300 animals were collected in the area around town of Sisak, where the outbreak had occurred. Bacteria had been successfully isolated only from three mice of the *Apodemus* species (*A. agrarius*) and in one hare (*Lepus europeus*) (Table 2) [38]. Even in next epidemic wave in 1967–1968, the assumed source of infection for humans was contact with infected hares. Epizooty among the hare population was detected at the time, in the tularemia-affected area (Figure 3). However, despite a strong increase in the population of small rodents, epizooty among them was never confirmed. From 58 collected small mammal species, *F. tularensis* was isolated only from one rat (*Ratus ratus*). Hence, according to the authors, this is not a significant result, since rats are an irrelevant species in an epizootiological and epidemiological sense. Because rats are cannibalistic animals, they are the last link in the chain of infection, therefore their role as source of infection is negligible [41].

Borčić and his group performed the most extensive five-year epizootic study in the early 1970s (1970–1975). The authors started from the hypothesis that the epizooty of tularemia preceded epidemic outbreaks, especially since 1967, and the increase in density of a small rodent’s population was determined. One of the main objectives of his study was to explain the reasons behind the tularemia outbreaks in Middle Posavina related to ecological characteristics. For that purpose, overall, 2428 small mammals were collected. The majority of the collected specimens were identified as the common vole (*Microtus arvalis*), yellow-necked mouse (*Apodemus flavicollis*), striped field mouse (*Apodemus agrarius*) and wood mouse (*Apodemus sylvaticus*). The common vole was indicated as a dominant species among the small rodent population in the investigated area. The endemic region was described as a meadow–field type of ecosystem with the common vole as the key feature in the epizooty of tularemia. The high reproductive capability of the common vole and the its susceptibility to *Francisella* infection further support this notion [55]. In this large study, more than two hundred animals were collected in two locations shortly before and immediately after the epidemic outbreak in 1974. Shortly before the outbreak, animals were collected in the wild around the town of Sisak, in the region of Middle Posavina. Upon the outbreak occurrence, collection was performed in a village near the town Sisak, in which human tularemia occurred. In attempts to isolate *F. tularensis* from wildlife, the authors performed the mouse bioassay method. Samples of liver and spleen were analysed in the pools of six animals from the same species and resuspended in saline solution. Suspension was given intraperitoneally to laboratory mice. Upon 10 days of incubation, only dead animals were further analysed. Spleen, liver and blood were inoculated into agar plates, containing blood, cysteine and glucose. Strains were identified by slide agglutination method with specific antiserum. Twelve isolates of *F. tularensis* were detected in four different mammal species, in these two collection sites (Table 3; Figure 3). Authors did not define isolated strains. Isolation of *Francisella tularensis* was a concrete evidence that small mammals play an important role as reservoirs of the infection and transmission to humans [56].

In parallel, to specify the role of arthropods as vectors of tularemia in the region of Middle Posavina, the same group started a nine-year survey. From 1970 until 1978, they collected more than 43,000 ticks belonging to the species *Ixodes ricinus*, *Dermacentor marginatus* and *Dermacentor reticulatus* at various locations in the region of Middle Posavina (Table 4). Samples of sorted ticks’ were processed in pools of 10–50 pieces together. The mouse bioassay method was performed for francisella isolation. *F. tularensis* was isolated only from one pool of *D. reticulatus* (Figure 2) [57]. Infected ticks were collected in wild nature, around the town of Sisak. This was the same location where *F. tularensis* had been isolated from animals in 1974.

Due to low incidence of the outbreaks of tularemia, the interest for investigating the epizooty of tularemia in Croatia decreased gradually. For many years, there has not been any data about the prevalence of tularemia among animals. The advent of molecular methods nowadays simplifies detection of *Francisella* in wildlife and maximizes accuracy of the results. Tadin and colleagues, showed the epizootic survey of tularemia between 2003 and 2011, with the aim of detecting a wide range of zoonotic agents among small mammals. Based on the polymerase chain reaction (PCR) detection method, two field mice (*A. agrarius*) trapped in the locality of Draganić in central Croatia, were found positive for *F. tularensis* (Figure 3) [58]. The Croatian Ministry of Agriculture started a program in 2016 for the determination of the prevalence of *F. tularensis* among animals in the Republic of Croatia [51]. Samples of small rodents were trapped during a two-year period (2014–2016) in localities in the middle course of the Sava river (Middle Posavina) and along the Drava river. The focus was directed on the localities situated near the Sava river, because it is known as the natural focus for *Francisella*, but also for other diverse rodentborne zoonotic pathogens, such as *Leptospira*, *Hantaviruses* and *Babesia* [58]. As a part of the mentioned program, in 2018, Mihelčić and colleagues, using quantitative real-time PCR tests, reported three positive mice from the *Apodemus* species found in the locality of Lipovljani, Middle Posavina (Figure 3), confirming that, after decades, this might remain the natural focus of *Francisella* species in Croatia [59].

## 4. Conclusions and Future Perspectives

Although rarely fatal and sporadic in occurrence, tularemia as a disease has been described in Croatia since 1952. Hence, epidemiologic and epizootic data about this zoonosis has not been consistent throughout the years. The most extensive studies were performed in 1960s and 1970s in parallel with epidemic outbreaks at time. It is a seasonal disease that occurs during autumn and winter. The findings presented in this review support the notion that the pathogen resides in specific ecological conditions [60,61]. In Croatia, tularemia has a long-term persistency in the area of the middle course of the Sava river, known as the natural focus of tularemia in Croatia. Five reported outbreaks in the second half of the 20th century and results given by the tularin test screening support this notion. The ecological specificity of this natural focus is the meadow–field type. Although contact with hares was presumed as the main source of infection for humans, epizootic studies revealed that the key factor for the epidemic outbreaks is fluctuation in the rodent population and occasional rodent abundance. The common vole was determined to play an important role in the epizooty of the disease in this area, although *Francisella* has also been isolated from other small mammals, such as yellow-necked mice, striped field mice, common shrew, bicolored shrew and hares. In addition, isolation of *Francisella* from ticks of *Dermacentor* species reveals that this arthropod, as a reservoir, may contribute to transmission of the disease.

In the past two decades, human outbreaks have abated, but the disease is still present, with sporadic occurrence. More than twenty reported tularemia cases in the past eight years confirm this hypothesis. Since concrete epidemiological data are not available, it is difficult to draw some concrete conclusions about nature of the disease in Croatia. Different forms of the disease have been reported in Croatia, with ulceloglandular, oropharyngeal and typhoid as predominant. Most of these tularemia cases could be connected to contact with contaminated water or to the consumption of contaminated food. Only in one case report from the second half of the 1990s were tick and mosquito bites the presumed sources of infection.

Tularemia is a zoonosis of complex epidemiology, but knowledge about the dominant mode of the transmission, the reservoir hosts and the vectors in nature are still incomplete. The historical data from the second half of the 20th century showed the predominance of the terrestrial cycle in the region of Middle Posavina, which is comparable with the ecology of *Francisella* in Central European Countries. In the recent years, the ecology of tularemia in the environment is under constant change, due to climate change, global warming and human and animal migrations [24]. On the other hand, a novel trend in the epidemiology of tularemia in Europe highlights the importance of the aquatic environment as a niche for the maintenance of *Francisella* in nature. Since there is no recent data for Croatia available, these facts should be taken into concern in the determination of infection sources and in planning future investigation surveys.

Furthermore, for Croatia to catch up and keep pace with the rest of Europe, movements forward in tularemia research are urgent. First, to determine the source of the infection for each reported case of human tularemia. Second, to isolate the bacteria from the source of the infection and to determinate the prevalent species. Third, to perform molecular characterization of Francisella isolates, from animals and humans, in order to complete the phylogenic map of *F. tularensis* subsp. *holarctica* in Europe. Novel molecular tools such as whole-genome sequencing assay and whole genome single nucleotide polymorphism (SNP) analysis have been described [35,61]. These assays allow characterization and discrimination of the strains, thereby defining population structure. Implementation of these tools, identification and definition of major clades present in Croatia would lead to better understanding of *Francisella* biodiversity, and enhance further epidemic investigations of tularemia in this country Definition of population structure would also elucidate movements of tularemia in the past, crossing the Croatian territory. Finally, until recently, only human cases of tularemia have been reported. To improve the monitoring of this disease, preventing a risk for human exposure, sustained research concerning epidemiological data and epizooty of tularemia is necessary. This should include research of other animals and vectors. In addition, all previous and current epizootic surveys have been focused only on the Croatian inland, relating to the outbreak occurrence. Future perspectives in the research should involve a larger geographical area, especially a coastal part of the country.

## Figures and Tables

**Figure 1 microorganisms-08-00721-f001:**
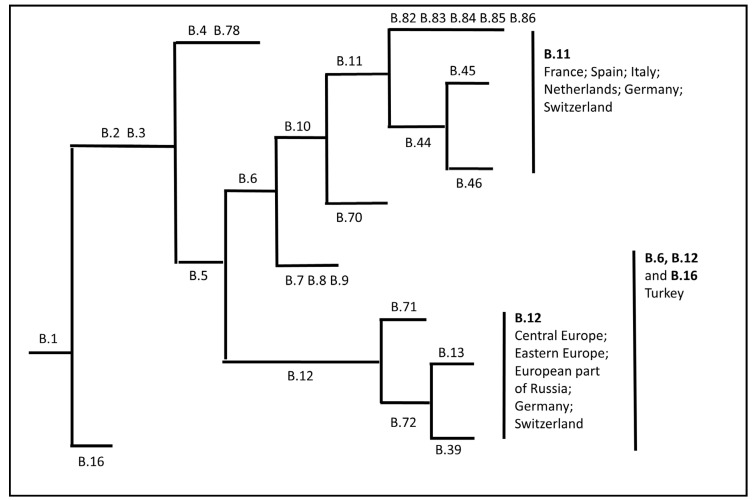
Schematic diagram of phylogeny of *F. tularensis* subsp. *holarctica* according to canonical single nucleotide polymorphism assay (canSNP) and its distribution across the Europe. Assay definition of each branch in the phylogeny. Branch length does not represent evolutionary distance and it is not scaled. Diagram was demonstrated according to the following references: Vogler et al. [35], Kevin et al. [30], Pilo al. [14] and Dwibedi et al. [27].

**Figure 2 microorganisms-08-00721-f002:**
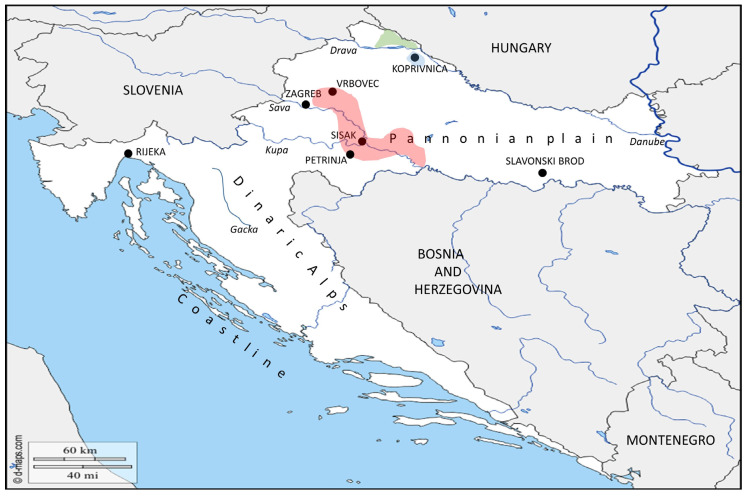
Natural foci of tularemia in Croatia. Three main natural foci of tularemia in Croatia according to Borčić’s 1974 study were determined. The greatest natural foci were situated along the middle course of the river Sava, in the region called Middle Posavina (red colour), and in two smaller areas located in Međimurje, northeastern Croatia (green colour) and around the city of Koprivnica (blue colour), nearby river Drava. Names of the cities and countries are labelled with upper case letters. Names of the three ecological regions are labelled with lower case letters. Names of the rivers are labelled with lower case letters and italic. (Blank maps adapted to source: https://d-maps.com/carte.php?num_car=5352&lang=en.

**Figure 3 microorganisms-08-00721-f003:**
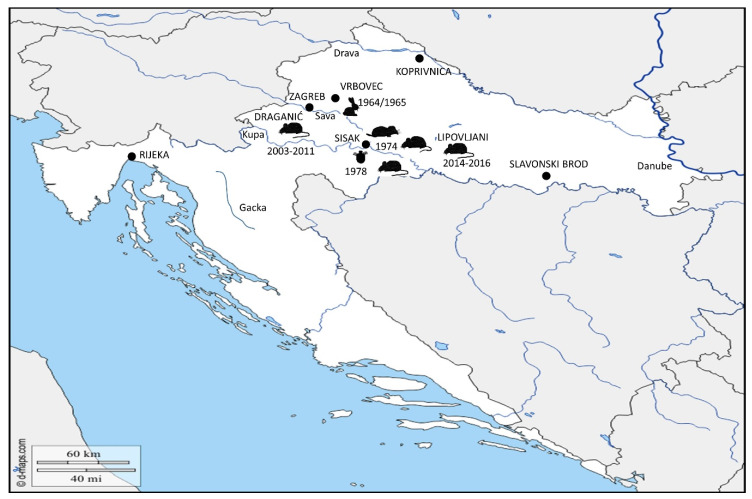
Epizooty of *F. tularensis* throughout the years in Croatia. Epizooty among the hare population (
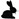
) was detected in 1964–1965 in localities around town Vrbovec. In 1974, the bacterium *Francisella* was detected in four different mammal species around town of Sisak: common vole (
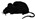
), stripped field mouse (
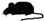
), common shrew and bicolored shrew (
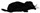
). In the same area, in 1978, the bacterium was isolated from *D. reticulatus* ticks (
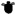
). Recent epizootic surveys reveal two mice positive for *Francisella* found in the locality of Draganić (2003–2011), and three positive mice collected on the locality of Lipovljani (2014–2016). Names of the cities and localities are labelled with upper case letters. Names of the rivers are labelled with lower case letters and italic. (Blank map adapted to source: https://d-maps.com/carte.php?num_car=5352&lang=en.

**Table 1 microorganisms-08-00721-t001:** Number of confirmed tularemia cases in Croatia 2012–2018, reported to the European Centre for Disease Prevention and Control (ECDC).

Year	Number of Reported Cases
2012	1
2013	2
2014	2
2015	13
2016	2
2017	3
2018	0
**Total**	**23**

**Table 2 microorganisms-08-00721-t002:** Epizootic survey from Heneberg’s research group upon epidemic outbreak in 1964–1965. Animals were collected in the affected area around town of Sisak.

Animal Species	Number of Collected Animals	Number of *F. tularensis* Positive Animals
*Apodemus flavicollis* (yellow-necked mouse)	3	0
*Apodemus agrarius* (striped field mouse)	87	3
*Apodemus sylvaticus* (wood mouse)	140	0
*Microtus arvalis* (common vole)	3	0
*Myodes* (*Clethryonomys*) *glareolus* (bank vole)	3	0
*Mus musculus* (house mouse)	2	0
*Crocidura leucodon* (bicolored shrew)	20	0
*Sorex araneus* (common shrew)	15	0
*Lepus europeus* (European brown hare)	10	1
*Vulpes vulpes* (red fox)	1	0
*Erinaceus* (hedgehog)	1	0
Total	285	4

**Table 3 microorganisms-08-00721-t003:** Epizootic survey performed by Borčić’s research group in 1974. Animals were collected at two locations in region of Middle Posavina, around town of Sisak.

Animal Species	Number of Tested Animals	Number of *F. tularensis* Isolates
*Microtus arvalis* (common vole)	87	4
*Apodemus agrarius* (striped field mouse)	70	4
*Apodemus sylvaticus* (wood mouse) and *Apodemus flavicollis* (yellow-necked mouse)	10	0
*Myodes* (*Clethryonomys*) *glareolus* (bank vole)	7	0
*Sorex araneus* (common shrew)	9	1
*Crocidura leucodon* (bicolored shrew)	22	3
*Mus musculus* (house mouse)	3	0
**Total**	**208**	**12**

**Table 4 microorganisms-08-00721-t004:** *Ixodid* ticks collected by Borčić and his group in the period from 1970 to 1978 in the region of Middle Posavina. Altogether, 43,532 ticks were collected in the region of Middle Posavina. The table below shows number of collected samples per species and per year, as well as total number. Ratio of collected single species is expressed as a percentage of total collected samples. *F. tularensis* was isolated only from one pool of *D. reticulatus*.

Species	Year									Total	%
	1970	1971	1972	1973	1974	1975	1976	1977	1978		
*Ixodes ricinus*	402	2280	235	84	19	95	9	33	69	3226	7.4
*Dermacentor reticulatus*	0	1237	982	1831	579	2788	3177	2415	2261	15,270	35.1
*Dermacentor marginatus*	0	488	2585	5385	4965	4432	3194	2637	1270	24,960	57.3
Undefined	0	0	0	0	0	0	0	0	76	76	0.2
**Total**	**402**	**4005**	**3802**	**7300**	**5563**	**7315**	**6380**	**5085**	**3676**	**43,532**	**100.00**

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
