# Peer review of "Epidemiologic and Epizootic Data of Tularemia in the Past and in the Recent History in Croatia"

_microorganisms, 2020, doi:10.3390/microorganisms8050721_

Round 1

Reviewer 1 Report

Thank you to the authors for this very interesting review of tularemia in Croatia. I have a few comments.

  • Lines 45-47: It would be useful to refer to Figure 1 to provide context.
  • Somewhere in the introduction it would be helpful to briefly describe the different forms of the disease
  • The discussion on clades and sub-clades is difficult to follow without reference to a phylogenetic tree. Is it possible for the authors to include a relevant tree?
  • When talking about the different outbreaks, it would be helpful to refer to the maps in the figures
  • Figure 1: it is difficult to make out the foci because of the scale of the map. What could be helpful is to leave the detail off of this map and provide an inset map at a finer scale that includes the detail you want to show, i.e. the foci and rivers. This could also provide an opportunity to show the different eco-regions mentioned at the beginning of the introduction.
  • Lines 197-199: 'typhoid form' is repeated twice, please delete one of them
  • Lines 204-215, including Table 1: Why do the number of cases differ between the ECDC data and the Croatian Agency for Agriculture and Food data? Does the latter just refer to cases that occurred in the 'continental part of the country'? If so, please make this more clear
  • Table 1: to me 2015 looks like an outbreak year. Is there any data around where cases came from? And if so, were there any geographical hotspots?
  • Lines 226-228: when it is stated that epizooty was never confirmed in small rodents, does this mean that they were tested and found negative? If so, please state this.
  • Epizooty section: it would be useful to provide some information about the tests that were used to survey wildlife in this section, much like what has been done in the 'epidemic investigations' section. My understanding is that F. tularensis can be difficult to isolate, so some discussion about how the introduction of molecular tests (which are likely more sensitive) could be beneficial to investigating the ecology of F. tularensis in wildlife hosts, would also be good.
  • Figure 2: This 'tick' icon does not show up in the legend

Author Response

We appreciate the constructive comments and all the issues have been addressed. 

Response to Reviewer 1 Comments

Point 1: Lines 45-47: it would be useful to refer to Figure 1 to provide context.

Response 1: Thank you for Your comment. Figure 1 is now Figure 2. Previous Lines 45-47 are now lines 52-54. Figure 2 is reffered in the text at line 54.

Point 2: Somewhere in the introduction it would be helpful to briefly describe the different forms of disease.

Response 2: Thank you for Your comment. Route of transmission and consequent different forms of disease are described and inserted in the main text, stariting at line 36. Paragraph shall read as follows: „Disease is transmitted by direct contact with infected animals or vectors, or indirectly. Indirect transmission includes consumption of food or water contaminated by animal excrement or carcasses. Consumption of contaminated water or food is the most frequently reported infection route in Europe. Then follows contact with infected animals and arthropod bites. Clinical manifestations of tularemia in humans depends on route of transmission. If transfer occurs by inoculation of bacteria via arthropod vector, glandular and ulceloglandular form of disease appear. After ingestion of bacteria by contaminated food or water oropharyngeal tularemia develops. Other forms of disease include occuloglandular and the most severe typhoid“.

Point 3: The discussion on clades and sub-clades is difficult to follow without reference to a phylogenetic tree. Is it possible for the authors to include a relevant tree?

Response 3: Thank you for Your comment. To make easier for readers to follow clades and subclades dispersal across the Europe, of F. tularensis subsp. holarctica, a schematic diagram of phylogenetic tree is introduced in the article as Figure 1, with the explanation in figure legend. Figure 1 is inserted in the main text at line 110, close to it's first citation, at line 98. Data in the provided sheme are assembled from the publications reffered in figure legend. In addition, to define preciously subclade B.Br. FTNF002-00, in the line 101, instead of „B.Br. FTNF002-00“ in the brackets, words „matches subclade“ are inserted. This sentence shall read as follows: Isolates belong to the Branch 11, which matches subclade B.Br. FTNF002-00, comprising B.45 and B.46 clades.

Point 4: When talking about the different outbreaks, it would be helpfull to refer to the maps in figures.

Response 4: Thank you for Your comment. Human outbreaks mentioned in the text are refered to Figure 2, in order to vizualize locations of the outbreaks occurences. Figure 2 has been reffered at line 125, line 160, line 169, line 210 and line 214.

Point 5: Figure 1: It is difficult to make out the foci because of the scale of the map. What could be helpful is to leave the detail off of this map and provide an inset map at a finer scale that includes the detail you want to show, i.e. the foci and rivers. This could also provide an opportunity to show the different eco-regions mentioned at the beginning of the introduction.

 Response 5: Thank you for Your comment. Previous Figure 1 is now Figure 2, due to changes in the manuscript. Figure 2 has been inserted in the main text at line 128., close to it's first citation (at line 125). It has been modified and presented in finer scale. Towns, rivers and natural foci shown on are now larger and easily seen and read. Ecological regions mentioned at the beginning of introduction: Coastline, Dinaric Alps and Pannonian plain have also been illustrated. Consequently, Figure legend of Figure 2 has been modified, according to modifications made in the map, line 130, and shall read as follows: „Figure 2. Natural foci of tularemia in Croatia. Three main natural foci of tularemia in Croatia according to Borčić 1974 were determined. The greatest natural foci situated along the middle course of the river Sava, in region called Middle Posavina (red colour); two smaller areas located on North-eastern Croatia, Međimurje (green colour) and around city of Koprivnica (blue colour), nearby river Drava. Names of the cities and countries are labeled with upper case letters. Names of the three ecological regions are labeled with lower case letters. Names of the rivers are labeled with lower case letters and italic. (Blank maps adapted to source: https://d-maps.com/carte.php?num_car=2172&lang=en). In addition, at line 194, Figure 1 has been now refered as Figure 2.

Point 6: Lines 197-199­: 'Typhoid form' is repeated twice.

Response 6: We appriciate Your observation. The term 'Typhoid form' is deleted. The sentence in line 221 shall read as follows: The disease was manifested with remittent fever and tularemia was confirmed serologically.

Point 7: Lines 204-215, including table 1: Why do the number of cases differ between the ECDC data and the Croatian agency for agriculture and food data?

Response 7: Thank you for Your comment. It may be because Croatian agency for agriculture and food speaking about food transmited outbreaks in 2015, reffer to one tularemia outbreak with 5 detected cases. We have noted this In line 236, adding words 'transmitted by food'. The sentence shall read as follows: „The Croatian Agency of Agriculture and Food reported one tularemia outbreak, transmitted by food, at 2015, with five confirmed cases reported to the Croatian Institute of Public Health“. Other detailed information about this outbreak, as some concrete epidemiological data or geographical region where outbreak occured, haven't been provided. Since there are no data available for the rest of the cases of tularemia reported to ECDC (alltogether 13 cases), we can only suppose that the other cases are not related wit the mentioned outbreak, and that they were not transmitted by consumption of contaminated food.

Point 8: Table 1: to me 2015 looks like an outbreak year. Is there any data around where cases came from?And if so, where there any geographical hotspots?

Response 8: Thank you for Your comment. Unfortunately, there are no case reports or data provided for this 13 confirmed tularemia cases. We presume that, tularemia was just diagnosed, without any epidemiological research.

Point 9: Lines 226-228: when it is stated that epizooty was never confirmed in small rodents, does this mean that they were tested and found negative? If so, please state this.

Response 9: We appriciate Your observation .In the manuscript, starting with line 251, we sought to clarify, why epizooty of infected animals was not confirmed. Paragraph shall now read as follows: „From 58 collected small mammal species, F. tularensis was isolated only from one rat (Ratus ratus). Hense, according to the authors, this is not significant result, since rats are irrelevant sepecies in epizootiological and epidemiological sense. Because rats are cannibalistic animals, they are last link in the chain of infection, thereby their role as source of infection is negligible.“

Point 10: Epizooty section: It would be useful to provide some information about the tests that were used to survey wildlife in this section, much like what has been done in the 'epidemic investigation' section. My understanding is that F. tularensis can be difficult to isolate, so some discussion about how the introduction of the moleculatr tests (which are likely more sensitive) could be benificial to investigating the ecology of F. tularensis in wildlife hosts, would also be good.

Response 10: Thank you for the comment and reccomendattion. In their atempts to isolate F. tularemsis in their wildlife surveys in the second half of 20th century, authors performed mouse bioassay method. The descriprion of the method is inserted in the text, starting at line 285. Text shall read as follows: „In attempts to isolate F. tularensis from wildlife, authors performed mouse bioassay method. Samples of liver and spleen were analyzed in the pools of six animals together, from the same species, and resuspended in saline solution. Suspension was given intraperitoneally to laboratory mice. Upon 10 days of incubation, only dead animals were further analysed. Spleen, liver and blood were inoculated into agar plates, containing blood, cysteine and glucose. Strains were identified by slide agglutination method with specific antiserum. Twelve isolates of F. tularensis were detected in 4 different mammal species, in these two collection sites (Table 3; Figure 3). Authors do not define isolated strains. Isolation of Francisella tularensis was a concrete evidence that small mammals play an important role as reservoirs of the infection and transmission to humans.“

The same method was performed in ticks' investigation survey. Thereby, a sentence „Mouse bioassay method was performed for francisela isolation“ has been added in the text at line 302.

Lines 314-315, a sentence „Recent epizooty surveys are based on molecular tests. Implementation of molecular tests ease detection of Francisella in wildlife maximizing the accurancy of the results“ has been added in the text.

At Line 317 was explained that Polymerase chain reaction (PCR) was used in survey performed by Tadin and collegues from 2003-2011. Sentence shall read as follows: „Based on polymerase chain reaction (PCR) detection method, two field mice (A. agrarius) trapped in the locality of Draganić in central Croatia, were found positive on F. tularensis (Figure 3).“

At line 325-328, a notification that quantitatice real time PCR was performed in detection of F. tularensis by Mihelčić and collegues, 2018 was inserted into the text. Sentence shall read as follows: „As a part of the mentioned program, in 2018 Mihelčić and colleagues, using quantitative real time PCR tests, reported three positive mice of Apodemus species found in the locality of Lipovljani, Middle Posavina (Figure 3), confirming that after decades this might remain the natural foci of Francisella species in Croatia.“

In section Conclusions and future perspectives, we have discussed benefits and necessity for implementation of novel molecular based analysis as whole - genome sequencing assay and whole genome single nucleotide polymorphism (SNP) analyses. Whole paragraph, starting at line 367, shall read as follows: „Third, to perform molecular characterization of Francisella isolates, from animals and humans, in order to complete the phylogenic map of F. tularensis subsp. holarctica in Europe. Novel molecular tools as whole - genome sequencing assay and whole genome single nucleotide polymorphism (SNP) analysis have been described. These assays allow characterization and discrimination of the strains, defining thereby population structure. Implementation of these tools, identification and definition of major clades present in Croatia would leed to better understanding of Francisella biodiversity, and enhance furher epidemic investigations of tularemia in this country. Definition of population structure would also elucidate movements of tularemia in the past, crossing the Croatian territory.

Point 11: Figure 2: This 'tick' icon does not show up in the legend.

Response 11: Thank You for your notification. The „tick icon“ has been inserted into the brackets.

Reviewer 2 Report

This is a very through review of the literature for tularemia.  I commend the authors for the extensiveness of this review. 

It is well written for the most part.  There are only  a few terms or style or writing that would be more compatible with scientific English writing.  Here are some examples:

p. 2, L 58.  should this not be river course rather that river curse as written?

P. 2, L 64. I suggest deleting the sentence with negligible.  It is not needed and is distracting.

P.3, line 101.  I would delete the word "clonal" in this sentence as it just is not the appropriate term and the sentence is accurate without it.

P. 3 line 122. The sentence refers to vaccinated persons remain sensitive to tularin. Was there an actual tularemia vaccine for people?  Or did hyou mean if they were given sub Q turlarin they remained positive for the tularin test?

p. 3 L 137 - 138.  the sentence " they were in contact once in their lifetime with F. tularensis"  I am not sure what you mean here.  Did they have the disease? or where they exposed to an infected animal or other source of F. tularensis?? Please clarify.

Otherwise, I think the writing is fairly clear. 

One question I would have.  In the Western U.S. , tularemia can be an occupational infection from handling sheep.  They become chronic infected with F tularensis and if the animal gets a cut or otherwise bleed onto the wool, shepards and sheep shearers can get infected -- mainly ulceroglandular.  Is that seen in Croatia"

Author Response

We appriciate the reviwer comments, and all the issues have been addressed.

Response to Reviewer 2 Comments

Point 1: p.2, L. 58. Should this not be river course, rather than river curse as written.

Response 1: We appriciate your observation. Line 58 is now line 65 in the text. The word ‘curse’ has been corrected into ‘course’, page 2, line 65.

Point 2: p.2, L. 64. I suggest deleting the sentence with negligible. Iti s not needed and is distracting.

Response 2: Thank You for your comment. Line 64 is now line 65 in the text. The sentence ‘Tularemia in wild animals cannot be negligible as well' has been deleted, page 2, line 70-71.

Point 3: p.3, L. 101. I would delete the word „clonal“ in this sentence as it just is not tha appropriate term and the sentence is accurate without it.

Response 3: Thank You for your comment. Line 101 is now line 108 in the text. The word ‘clonal’ has been deleted, page 3, line 108.

Point 4: p.3, L. 122. The sentence refers to vaccinated persons remain sensitive to tularin. Was there an actual tularemia vaccine for people? Or did you mean if they were given sub Q tularin they remained positive for the tularin test?

Response 4: We appriciate Your observation. In the reffered aricle of Heneberg et colleagues , authors describe how they tested specificity of the tularin test. Live vaccine, was administred to two volunteers for experimental purpose, before administration of tularin. Details about vaccine are not provided, or referred, it has been just mentioned that vaccine originated from former Soviet Union. Line 149-150, sentence has been modified and shall read as follows: „After the intracutaneous administration of tularin, previously infected persons or persons which were vaccinated with live vaccine for the experimental purpose, developed hypersensitivity.“

Point 5: p.3, L. 137-138. The sentence “they were in contact once in their life with F. tularensis“. I am not sure what you mean here. Did they have the disease? Or where they exposed to an infected animal or other source of F. tularensis?

Response 5: Thank You for your comment. Lines 137 is now 168. For better understanding, paragraph has been modified, starting with line163 and shall read as follows: „For retrograde detection of tularemia cases among inhabitants, which had not been reported previously in the designated area, the tularin test was performed. Alltogether, 623 healty persons were involved in the survey, 27 of them (4,3%) were positive on the tularin test. According to the authors, they might became infected with tularemia in one of the previous outbreaks and remained positive on tularin test.“

Unfortunately, authors do not provide any epidemiological information whether they had some clinical signs previously or not. Authors also do not provide information about presumed sources of infection for them.

Point 6: In the Western U.S. , tularemia can be an occupational infection from handling sheep. They become chronic infected with F tularensis and if the animal gets a cut or otherwise bleed onto the wool, shepards and sheep shearers can get infected -- mainly ulceroglandular. Is that seen in Croatia"

Response 6: That is very interesting report. Unfortunately, there are no dana available about tularemia in domestic animals in Croatia. Neither we have reports about tularemia in sheeps. There's also no case report indicating that shepards and sheep shearers got the infection through the sheeps.

Round 2

Reviewer 1 Report

Thank you for addressing my previous comments and for modifying/adding the figures as suggested. You have definitely improved the manuscript.

In regards to figure 1, would it be possible to add onto the figure (perhaps on the right hand side) the geographic regions represented in each of the major clades. I know you go through this in the text, but having it on the actually figure would make it easier to visualise.

The language structures needs some improvements, examples include:

  • Line 52-53 - change the end of the sentence to 'taken into consideration.'
  • Line 87 - change to 'countries neighbouring Croatia'
  • Line 175 - add a 'h' to healthy
  • Line 217 - does "flees" refer to fleas or flies?

Author Response

Response to Reviewer 1 Comments

Point 1: In regards to figure 1, would it be possible to add onto the figure (perhaps on the right hand side) the geographic regions represented in each of the major clades. I know you go through this in the text, but having it on the actually figure would make it easier to visualise.

Response 1: Thank you for your comment. On the right side of the Figure 1, geographical distribution F. tularensis subsp holarctica major clades, across European continent, has been noted. Figure 1 has been inserted in the main text at line 110. Title of the Figure legend, line 112 has also been modified and shall read as follows: “ Figure 1. Schematic diagram of phylogeny of F. tularensis subsp. holarctica according to canonical single nucleotide polymorphism assay (canSNP) and its distribution across the Europe.

At line 102, in accordance with changes in Figure 1, to further specify the geographical distribution of some clades in Europe, “Germany and Switzerland” have been added in the sentence and referred. The sentence shall read as follows:” Subclade B. Br. FTNF002-00 has been reported along with France, Spain, Italy, Netherlands, Germany and Switzerland, while B.12 clade and its subclade B.13 are mostly distributed in Central, Eastern Europe and the European part of Russia“

Point 2: The language structures needs some improvements examples include:

Line 52-53 - change the end of the sentence to 'taken into consideration.'

Line 87 - change to 'countries neighbouring Croatia'

Line 175 - add a 'h' to healthy

Line 217 - does "flees" refer to fleas or flies?

Response 2: Thank you for your suggestion.

Line 52-53- The end of the sentence has been changed to 'taken into consideration.' The sentence shall read as follows: “While investigating tularemia in Croatia, its geographical location should definitely be taken into consideration.”

Line 87 – part of the sentence has been changed to 'countries neighbouring Croatia'. The sentence shall read as follows: “Due to the frequency of reported outbreaks of water-borne tularemia in countries neighbouring Croatia, located in the south-east of Europe (Kosovo, Bulgaria, North Macedonia, Turkey), the existence of an aquatic cycle cannot be excluded even on the Croatian geographic territory.

Line 163 – a letter 'h' has been added to word “healthy”.

Line 205 – word "flees" refer and has been changed to flies.
